# Approaches for Disseminating Environmental Research Findings to Navajo Communities

**DOI:** 10.3390/ijerph18136753

**Published:** 2021-06-23

**Authors:** Tommy Rock, Lindsey Jones, Jani C. Ingram

**Affiliations:** 1School of Earth Science and Environmental Sustainability, Northern Arizona University, Flagstaff, AZ 86011, USA; rockt92@gmail.com (T.R.); lmj53@nau.edu (L.J.); 2Department of Chemistry & Biochemistry, Northern Arizona University, Flagstaff, AZ 86011, USA

**Keywords:** environmental research, dissemination, community collaboration

## Abstract

We report the use of three different dissemination approaches for providing environmental research results back to Navajo communities from different research projects. The objectives of the dissemination are to provide the results to the community, have a dialogue about the results, and learn more about the environmental concerns of the community for potential future research projects. The first approach utilizes radio announcements and flyers provided to the community announcing dissemination meetings specific to the research projects. The second approach is more collaborative, working with a grassroots organization to organize report-back meetings, as well as one-on-one discussions of the research project. The third approach involves the development of a booklet for distribution to communities along with an oral presentation at the regularly scheduled monthly community meetings to discuss the information. Overall, the second and third approaches are more effective than the first approach in terms of dissemination to a larger number of community members, as well as increased dialogue between the researchers and the communities.

## 1. Introduction

The Navajo Nation is the largest contiguous Native American reservation in the continental United States. Located within the Four Corners region of the American Southwest, its borders span 71,000 square kilometers across Arizona, New Mexico, and Utah. The Navajo Nation is recognized by the United States’ government as a sovereign nation, though the United States retains plenary power. The issue of sovereignty is crucial in understanding how researchers from the outside can work with American Indian tribes because the original treaties between the U.S. federal government and American Indians give American Indians quasi-sovereign control over their lives and reservations [1]. Permissions to collet research data (human, environmental, economic, etc.) must be reviewed and approved by tribal leaders before any work can be initiated. Each tribe has their own processes for gaining approvals; thus, researchers must acquaint themselves with these processes. On the Navajo Nation, researchers work with communities to gain Resolutions of Support. Additionally, researchers may need to gain Resolutions of Support from tribal health boards, the Navajo Historic Preservation Department, and other entities that may need to review the project. These Resolutions of Support must be secured before any research can begin.

The Navajo Reservation is divided into 110 tribal Chapters governed through five management Agencies: Chinle (14 Chapters), Crownpoint/Eastern (31 Chapters), Fort Defiance (27 Chapters), Shiprock (20 Chapters), Tuba City/Western (18 Chapters) [1]. Comparing the division of regions within the Navajo Reservation to the United States, the Agencies are similar to states, whereas the Chapters are similar to counties. The Navajo Nation is located within the Colorado Plateau region. From 1948 to 1956, uranium mining took place on the Colorado Plateau and the Navajo Reservation. Past activities resulted in an estimated 1200 mines sites the Navajo Reservation [2]; however, most of the mines were concentrated in the Four Corners area, the southeastern area near Church Rock, NM, and the southwestern region near Cameron, AZ.

While the link between environmental uranium exposure and health problems is still being determined, the perception of the Navajo people is that the uranium has poisoned people from their communities [3,4]. The Navajo’s perspective can be understood considering the many abandoned mines following the mining boom, general fear surrounding environmental uranium exposure and an increased risk of cancer, kidney disease, and other health problems, and their limited success in gaining compensation following the passage of the Radiation Exposure Compensation Act in 1990 [5]. The environmental analytical laboratory at Northern Arizona University (NAU) has been focused on working with Navajo communities to analyze environmental samples to provide a better understanding of uranium and other mine-related contaminant exposures to the Navajo people [6,7,8,9].

The objective of this paper is to report three dissemination approaches taken to provide environmental research results to Navajo communities. In working with communities on their environmental concerns, reporting back the results of the research in a timely, understandable way, to as many of the community members as possible, is critical for researchers to gain trust from the community, and more importantly, address the concerns of the community through collaborative interactions [10]. Each dissemination approach described in this paper was for a different project; however, all focused on environmental issues associated with mining activities on the Navajo Nation. Briefly, the first and second dissemination approaches were a series of community meetings on or near the Navajo Nation providing environmental findings. The first approach utilized radio announcements and flyers for advertising the meetings. The second approach utilized a partnership with a grassroots organization to advertise the meetings. The third dissemination approach was presentations of the research findings at regularly scheduled monthly community meetings. No advertising of the presentation of findings for the third approach was made prior to the meetings, since the dissemination was an agenda item within the monthly community meeting. Additional detail on each of these approaches is provided below.

## 2. Methods

Environmental research in the Navajo communities described in this paper was initiated with discussions of environmental issues between the researchers and Navajo community members prior to any research taking place. Two of the authors (Rock and Ingram) are members of the Navajo Nation, so they have familiarity with Navajo tribal culture. Although the research was being led by Navajo Nation tribal members, it was still very important for the researchers to work with specific communities to determine the issues of interest to the community, as well as how the project should be appropriately carried out. For all the research projects described in this paper, Resolutions of Support were gained to ensure the community was in agreement with the research plan [11]. Since the Navajo Nation is a sovereign entity, approvals for all research projects must be provided by the communities prior to any initiation of work [12]. A requirement of every research project that impacts the Navajo Nation is the dissemination of the research findings to the community prior to any publication of these findings to the larger scientific community. The dissemination approach is important to provide the research findings to the community in an effective manner. The purpose of this paper is to provide dissemination insights for other researchers planning to provide their research findings to Navajo stakeholders. It should be noted that a formal evaluation of the dissemination approach described in this paper was not done; the purpose of this paper is to share the lessons learned with other researchers interested in improving their dissemination activities.

## 3. Dissemination Approach One-Public Meetings

In collaboration with the Colorado Plateau Foundation and Tolani Lake Enterprises, a conference series entitled, “Restoring K’e: The Lessons Learned to Heal Our People, Our Land, and Our Waters from Uranium contamination” was organized on the Navajo Nation. The conferences were geared toward informing tribal communities about uranium contamination and exposure with presentations specific to their community concerns. The conferences took place in Shiprock, NM (northeastern Navajo), Sanders, AZ (southeastern Navajo), Cameron, AZ (southwestern Navajo), and Kayenta, AZ (northcentral Navajo). To promote the conferences, radio announcements, and flyers were provided to the community announcing dissemination meetings to provide results back to the community.

The conference organizers invited researchers from the University of New Mexico, Northern Arizona University, University of Arizona, and Dine’ College. The invitation was extended to the Indian Health Service medical providers, including doctors, as well as regulators (Federal and Navajo). Invitations were extended to representatives of the Navajo Nation Radiation Exposure Compensation Act (RECA). The regional tribes were also invited. Lastly, because environmental science is conducted mostly from a western perspective, Traditional Knowledge Holders were invited as well (Medicine Women and Men) to ensure that Traditional Ecological Knowledge was presented to conference attendees. Food was provided at the events through community collaboration.

The conferences were designed to appeal to adult and youth community members, as well as health care providers in uranium-effected communities. The presentations were given in English with translation to the Navajo language by Dr. Rock (one of the co-authors). The intended outcomes of these conferences were to increase awareness of the uranium contamination issues, and develop of water quality testing skills through hands-on demonstrations, and community empowerment.

The first conference, held in Shiprock, NM, focused on abandoned uranium mine issues, as well as the Gold King Mine spill. This spill occurred in August 2015 because of a breach to a holding pond at the Gold King Mine and results in over three million gallons of mine waste flowing down the Animas River, then pouring into the San Juan River that flows across the northern part of the Navajo Nation [13]. At this meeting, there were approximately 10 community members. The second conference, held in Sanders, AZ, focused on the Puerco-Little Colorado River Watershed on the uranium water quality project. In particular, an environmental justice project was done using community-based participatory research to investigate the public water system, which violated the maximum contaminate level for uranium over a 10 year period. At this meeting, only a few community members attended. It was interesting as one of the attendees was a community elder who was frustrated that attendance at the meeting was poor; he wanted an additional meeting because he felt the information provided to the community from this study was important. The elder even volunteered to help the researchers in informing the community. A follow-up meeting was organized for the Sanders community in collaboration with the elder was held later. The third conference, held in Cameron, AZ, focused on the open pit abandoned uranium mines in the region. This conference was the best attended of the three conferences, with approximately 25 community members present and more attending via a Facebook live stream. The fourth conference, held in Kayenta, AZ, focused on the abandoned uranium mining in northern Arizona and southern Utah. This conference was poorly attended, with approximately five community members present; we learned that there were other events occurring on the day of the conference that most likely resulted in the low attendance. It should be noted that the meetings that were poorly attended were the fault of the researchers for the poor advertisement of the meeting, and/or choosing times and dates for the meeting that were not optimal for the community.

## 4. Dissemination Approach Two-Partnership with Grass Roots Organization

The second dissemination approach was to provide information on a project centered in Cameron, Arizona, which is a small community on the Navajo Nation in Northern Arizona. Cameron is on the western border of the Navajo Nation. The small community has a population size of 885 using the 2010 Census with a chapter land size of 238,523 acres [14]. Past uranium mining in the Cameron community extends back to 1952 [15]. Cameron has 98 open pit abandoned uranium mines that have been documented by the United States Geological Survey [15]. Research in the Ingram Laboratory at Northern Arizona University has worked with the Cameron community to identify and assess uranium contamination of water, soil, plants, and livestock [7,16,17].

The second dissemination approach was somewhat similar to the first dissemination approach in that it was a series of four meetings held specifically to provide environmental research findings. Similar to the first approach, food was provided for the community members at each meeting. Presentations were also given in English with translation in the Navajo language by Dr. Rock (one of the co-authors). However, in the second dissemination approach, the dissemination was for a single community, Cameron. The second dissemination approach was supported by the Center for American Indian Resilience Project (CAIR) at Northern Arizona University. The CAIR Project was focused on working with the local grassroots organization on informing the local community members about different pathways of exposure to uranium. The project was a steppingstone related to Indigenous food contamination and getting the elders and Traditional Knowledge Holders involved in the research. The CAIR project was a collaboration between the researchers at Northern Arizona University and the Forgotten People, a non-profit grassroots organization that addresses environmental and social justice. The purpose of the CAIR Project was to work with a community organization, the Forgotten People, to raise awareness of the health hazards of uranium exposure and contamination. Additionally, the project’s aim was to transfer knowledge about traditional Indigenous food and water use in the Cameron community of the Navajo Nation to the researchers. The long-term goal of this work is to develop culturally appropriate policy using the Navajo Fundamental Laws addressing contaminated traditional food on the Navajo Nation [9].

This project obtained a signed Memorandum of Agreement (MOA) between the researcher (a member of the Navajo Nation), from Northern Arizona University, and the grassroots organization, the Forgotten People. The MOA is an example of trust-building that promotes successful collaboration with a grassroots organization on addressing environmental justice [5,18]. The MOA was unique to this dissemination approach. This collaborative partnership helped to bring community members from isolated rural areas to the community educational meetings.

The CAIR project included four meetings with the community members of the Cameron Chapter. For these meetings, the grassroots organization, the Forgotten People, arranged for the meetings to be held in a room in a local business building. The Forgotten People also chose the date and time for the meetings. The focus of the meetings was the health of the community in respect to the abandoned uranium mines in the area [19]. Previous studies have demonstrated a clear pathway of exposure to ill health from around abandoned uranium mines on the Navajo Nation [20]. These meetings discussed these pathways of exposure along with the critical need for avoidance of these areas, contamination of water and foods, as well as the urgent need for cleanup. The meetings were well attended, and the number of attendees increased from 15 to 30 community members from the first meeting to the last meeting. There was active discussion and interest, as well as a concern about institutional racism when it came to abandoned uranium mines on the Navajo Nation [21].

Due to the overwhelming interest of participants, the last meeting was a tour of the researchers’ laboratory at Northern Arizona University in Flagstaff, Arizona. This meeting included a series of presentations at the Native American Culture Center on the Northern Arizona University campus, as well as providing lunch. Three Navajo student researchers presented their research on environmental contamination research. Additionally, the community members were given a laboratory tour, and demonstrations of the chemical analyses were provided. At the end of the meeting, the community members and researchers met for a listening session on the environmental concerns of the community.

## 5. Dissemination Approach Three-Distribution of a Results Booklet

The third dissemination approach was to provide information on uranium and arsenic levels in unregulated water sources in the western region of the Navajo Nation and included twelve Navajo Chapters.

Uranium and arsenic levels were highest in the southwest portion of the study area. Temporal variability of uranium and arsenic concentrations was observed in a subset of wells, especially shallower hand-dug wells and hand pumps. The overall results were compiled into a booklet which was presented to western Navajo Nation Chapters included in the study, as well as the Navajo Department of Water Resources and the Navajo Nation Environmental Protection Agency [8].

This research combined physical and social sciences, which are critical to achieving solutions to environmental challenges. Field and chemistry work was essential to provide the data; however, social interactions, such as community presentations, were critical to making the data relevant. In other words, one would be ineffective without the other. The relationship between researchers and community members is also important to consider. This research focused on improving relations between the two groups and creating an open dialogue that solves problems. Understanding the steps and procedures that are needed to do research on the Navajo Nation was extremely important. The Resolutions of Support from the various Chapters helped to engender trust that this research was respectful of Navajo customs. This research provided much-needed insights into the quality of water available to the community members. These results can be useful to provide data for comparison of future water quality testing, for determining particularly problematic mining areas, and to determine the existence of possible natural sources of dissolved uranium and arsenic.

The third dissemination approach was different than approaches one and two. In the third approach, the dissemination process was two-fold. First, the project results were summarized for each Chapter. A booklet was constructed that had a map of the water wells tested for the Chapter on the left page of the booklet, and a summary of the uranium and arsenic results were provided using bar graphs on the right side of the booklet. Additionally, a short description of the conclusions is provided beneath the graphs. The booklet provided results for all 12 Chapters, along with a brief summary of the project written in the Navajo language. Researchers requested the Chapter leaders to be added to the monthly agenda to provide an in-person oral report on the results of the study. All requests were granted; an oral presentation was made at the Tuba City and Cameron Chapters’ monthly meetings, at the Tuba City and Cameron Grazing Permittee meetings, and at the quarterly Western Navajo Agency meeting. The attendance at these meetings varied; in all cases, the attendance was typical of other monthly or quarterly meetings. Booklets were handed out to community members present at the meetings, as well as leaving additional booklets for distribution to community members who could not attend the meeting. Additionally, booklets were sent to all 12 Chapters participating in the study. The oral dissemination was part of a monthly chapter or grazing meeting and quarterly agency meeting. Thus, it was a single agenda item within a larger meeting agenda. No advertisement for community members to attend was done specifically to the dissemination. The presentations were given in English with translation to the Navajo language by one of the Navajo leaders at the meeting. We found that providing community members both the booklet and presentation gave the community members an opportunity to ask questions to clarify the topic is beneficial for disseminating results [22]. The response from the communities was very positive, with requests for additional booklets for community members unable to attend the presentations. These dissemination activities could not have occurred without the help of Chapter officials who translated the presentations at the meetings in the Navajo language.

## 6. Conclusions

The lessons learned from disseminating the results of environmental research to Navajo communities were many. It was clear in comparing the three approaches described in this paper, the less personal approach to marketing dissemination meetings, such as flyers and radio announcements, is minimally useful for getting community members to attend meetings as described in Dissemination Approach One. It was clear from the Sanders elder that the information was important to the community, and he thought more community members would want to know about the study. We speculate that community folks have many activities in their lives, so the impersonal invitation to attend a meeting was not enough; other factors, such as time, place, and personal invitations, would have most likely increased attendance.

In Dissemination Approach Two, the partnership with the grassroots organization provided a more personal interaction with the community—with the grassroots members acting as a type of liaison between the researchers and the community. The MOA helped in establishing roles of responsibility for the researchers and the grassroots organization. This approach, although somewhat similar to Dissemination Approach One in that dissemination was provided in public meetings specifically focused on the research, was more effective. We speculate that the grassroots organization was key to persuading the community members that it was worth their time to attend the public meetings. Moreover, the grassroots organization chose the time and place for the meetings. Dissemination Approach Two resulted in higher attendance at the dissemination meetings, with attendance growing at each subsequent meeting—suggesting that word of mouth among the community members was also helping to market the public meetings. An important success in Dissemination Approach Two was having the community members tour the research lab at Northern Arizona University, which helped the community members understand the process of the environmental samples that were analyzed. The tour helped bridge an understanding of the time-consuming work that is required to analyze samples.

Dissemination Approach Three took advantage of presenting the results at a monthly Chapter meeting which many community members attend regularly. The unique aspect of this approach was to develop a booklet describing the results in a way that was easy for the community members to understand. Many community members commented on the ease by which they could find the water well of interest on the map and see what the uranium and arsenic results were for that well. They also liked being able to take the booklet home to show to other friends and family members. Finally, they asked several questions during the oral presentation of the results, which suggests that they understood the material presented. This is not always the case when researchers present their scientific results to non-scientific community members.

Through the lessons learned from the three dissemination approaches described here, the following are recommendations that should be considered when reporting research findings to Navajo and other tribal communities.

When planning a meeting specifically to report the research findings that are not connected to any other meeting or event in the community, it is important to make sure the date and time chosen for the meeting do not conflict with other events occurring in the community.Advertisement for the meeting is more successful if community members assist in recruiting the community to attend the meeting (such as was done in the second dissemination approach). Flyers, radio ads, and other non-personal invitations are also useful, but these approaches should be augmented with personal invitations from community members to other community members.Providing refreshments at the meeting is often expected from the community. The food and beverages do not need to be extravagant, but food is important in bringing people together.Presentations should be provided in a language that is understandable to the audience. In the case of all approaches, layman terms and translation to Navajo were important for communication.Partnerships with tribal community leaders are key to understanding community activities, such as monthly meetings and events that can be critical times to report back information, since the community members will already be attending these events. Being respectful of requests to be added to meeting agendas is very important, as well as sitting through the majority of the meeting to better understand the community.It is also important to be open to staying beyond the meeting to speak with community members one-on-one to answer questions that may not have been asked during the presentations. This also helps to build new relationships with community members.It is useful to provide hard copies of the research findings in layman’s terms for the community members present, as well as leaving copies with the tribal community leaders for distribution to community members who could not attend the meeting is greatly appreciated.

An important part of research in communities is providing the results to the community members as much of the research is grounded in concerns from the community. However, when researchers do not try to disseminate results in an understandable way to as many of the community members as possible, then dissemination is not achieved. This leaves communities with no outcomes from the research and leads to distrust of researchers. Critically important for dissemination is to work with the community to find approaches that will achieve success in providing the outcomes of the research to the community to address their concerns.

## Data Availability

Not applicable.

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
