# Peer review of "Approaches for Disseminating Environmental Research Findings to Navajo Communities"

_ijerph, 2021, doi:10.3390/ijerph18136753_

Round 1
Reviewer 1 Report
The introduction to the Navajo Nation is good but the introduction to the three approaches was not sufficient. What are the similarities among the approaches and what are the significant differences?
The methods section introduced the researchers but did not adequately describe Tribal sovereignty or the reasons for MOAs and resolution. A person who does not do Tribal work would not understand the significance of these issues from the method section. Also, the requirement to disseminate results is not well discussed. Is it a requirement of human subject’s protection or is it codified in Navajo Nation law? What data was used to draw conclusions? Was it a formal evaluation process, an external evaluator, or research reflections?
The discussion of the community partners was well described. However, details of the meetings for each approach were missing. Was it only in the first approach that food was provided? It would be important to discuss why it was significant to discuss providing food in a Tribal context. It was unclear if the three approaches were sequential or when they happened. Were they associated with different research projects?
There was inconsistency in identifying the number who attended each meeting in the three approaches. Again, no description of the evaluation of the effectiveness of each meeting, information dissemination method or community response was provided. In approach one did the researchers work with the elder identified? If not, what were the limitations to that follow up?
The third approach would benefit from a description of the steps taken to get invited to speak at Chapter meetings. In that approach the booklet was well described but the chapter meeting presentations were not. The manuscript says in the final paragraph that the results were provided at several meetings but not an exact number. The importance of translations was not discussed only mentioned.
The conclusion would benefit from a set of recommendations with a reiteration of the importance of each. This is especially needed for researchers who are not experienced with Tribal work.
Author Response
Response to Review 1
- Suggestion to add more detail in the comparison of the three dissemination approaches – similarities and differences
- We have added more detail in lines 68 to 77.
- The reviewer suggested we add information on tribal sovereignty.
- We have added this information in lines 33 to 43.
- The reviewer asked for more detail on the dissemination meetings themselves such as the number of attendee, if food was provided, etc.
- We have added detail in each of the sections describing the three approaches. We gave some similarity/differences in those descriptions to help the reader understand the contrasts of these approaches.
- The reviewer asked to clarify if the three different dissemination approaches were for three different projects or for the same project.
- In Lines 68-69, we have clarified that the approaches were used for three different environmental projects.
- The reviewer asked if the dissemination approaches were evaluated.
- In the methods section, we added a sentence at the end of the section stating that no formal evaluation was done (lines 94-96). We are reporting lessons learned.
- The reviewer commented that there was an inconsistency in reporting the number of attendees at the various meetings.
- We have provided estimated numbers of attendees for the meetings described.
- The reviewer asked about follow-up with the elder mentioned at a poorly attended meeting in Sanders.
- A sentence was added to lines 135-136 about an additional meeting that was organized with the assistance of the elder.
- The reviewer requested a set of recommendations based on the work presented in the manuscript.
- Recommendations were provided in lines 294-321 as a bulleted list.
Reviewer 2 Report
Strengths - This is a well-written manuscript that conveys different modes of data dissemination to a community partner. This paper is particularly relevant as Tribal nations develop their research agendas with universities, with researchers often unaware of appropriate data translation methods. These methods can be used as a model for working with Tribal nations.
Weaknesses - perhaps indicate that this is a qualitative observational study, as there is little numerical data and statistics to support your findings. Regardless, this is an important approach to be shared with scientific researchers. I'm also wondering if the researchers needed to obtain approval from the Navajo Nation Human Research Review Board and if this needs to be included in the paper to inform others of the approval process.
Author Response
Response to Review 2
- The reviewer asked about quantitative or numerical data.
- No quantitative or numerical data are presented as there was no formal evaluation. The intent of this manuscript is a reflection on the three dissemination approaches based on experiences of the authors.
- The reviewer asked if approval is needed by the Navajo Nation Human Research Review Board for the paper.
- All research that involved human subjects, such as a survey, was reviewed and approved by the Navajo Nation Human Research Review Board. The content of this manuscript is not human research, but it is a reflection of the experiences of the authors in dissemination of environmental science findings to Navajo communities.